# Screening of Cold-Adapted Strains and Its Effects on Physicochemical Properties and Microbiota Structure of Mushroom Residue Composting

Min Wang [1], Haoxin Lv [2], Liping Xu [3], Kun Zhang [3], Yan Mei [3], Shengtian Zhang [3], Ming Wu [4], Yifei Guan [1], Miao Zhang [1], Huili Pang [1,*] and Zhongfang Tan [1,*]

[1] Henan Key Laboratory of Ion Beam Bio-Engineering, School of Agricultural Science, Zhengzhou University, Zhengzhou 450000, China; wangmin509867@163.com (M.W.); guanyifei0904@163.com (Y.G.); miaozzu@163.com (M.Z.)
[2] School of Food Science and Technology, Henan University of Technology, Zhengzhou 450001, China; lvhaoxin0129@126.com
[3] Xining Vegetable Technical Service Center, Xining 810016, China; xnxuliping@163.com (L.X.); qhghgx@163.com (K.Z.); ylgever@163.com (Y.M.); stxn0125@163.com (S.Z.)
[4] Qinghai Special Fertilizer Co., Ltd., Xining 810005, China; 18397019694@163.com
* Correspondence: pang@zzu.edu.cn (H.P.); tzhongfang@zzu.edu.cn (Z.T.)

**Abstract:** Composting is an effective way to dispose of agricultural waste; however, its application is limited in the winter and in areas with low average annual temperatures. This study screened out a composite microbial agent (CMA) including *Bacillus* (*B.*) *cereus* QS7 and *B. pumilus* QM6 that could grow at 10–15 °C and investigated the effects of the CMA as an inoculant on the physicochemical parameters and microbial communities of compost made from mushroom residue mixed with sheep manure. The results showed that CMA inoculation prolonged the days of high temperature above 60 °C. When the ratio of mushroom residue to sheep manure was at 9:1, compost increased the accumulation of nitrogen, and reduced the electrical conductivity (EC). Under this ratio, the inoculation of CMA accelerated the degradation of organic matter (OM) (decreased by 12.22%) and total organic carbon (TOC) (decreased by 8.13%) and increased the germination rate and seed germination index (GI) to 100% and 106.86, respectively. Microbial community structure analysis showed that the relative abundance of *Flavobacterium* was 30.62% on day 15 and was higher after CMA inoculation in the high mushroom residue condition, which was higher than that of other groups, and the relative abundance of thermophilic fungi increased. This study demonstrated that CMA improves the quality and efficiency of mushroom residue and sheep manure composting, and it provides evidence to improve the efficiency of low-temperature composting.

**Keywords:** mushroom residue; sheep manure; composite microbial agent; physicochemical; microbial community structure



## 1. Introduction

As at least a third of the estimated 9.05 billion tons of agricultural waste produced globally each year is not disposed of in an environmentally safe manner [1], it is urgent to find more effective treatment methods. Sustainable management methods for agricultural waste include composting, landfilling, decomposition, etc. [2]. Composting is a common method for recycling agricultural waste, which can be converted into mature and a stable end product and used as substrate for plant cultivation [3–6]. It can not only stabilize organic matter (OM) and promote nutrient element recycling, but can also kill harmful organisms through high temperatures [1,7] and convert waste into stable, sanitary, and pollution-free materials [8,9]. However, composting requires a certain level of compost heap temperature. Low-temperature conditions may cause composting to fail to start properly

or to reach the high temperatures required to eliminate pathogenic microorganisms and harmful pollutants, posing technical challenges to the normal process of composting. To ensure the normal process of composting under low-temperature conditions, more and more researchers have studied heating methods for composting, such as biogas heating [10], electric heating, and cover cropping [11–13]. However, these methods require a continuous supply of external energy and are not energy efficient or economical for large-scale composting, and microbial inoculation has been widely used as another way to accelerate composting [13].

Microorganisms play an important role in composting. Microbial inoculants can accelerate the degradation of agricultural waste cheaply and easily in operation compared to other methods such as biogas heating and electric heating [14,15]. An increasing number of researchers have studied the effect of using microbial inoculants in compost. Lu et al. [9] showed that inoculation with compound straw-decomposing microbial agents (a combination of cellulolytic microbes, *Bacillus*, and actinomycetes) accelerated the increase in compost pile temperature, promoted the degradation of total organic carbon (TOC), and increased the stability of the fungal network. The inoculation of a compound bacterial agent (a combination of *Acinetobacter pittii*, *Bacillus subtilis* sub sp. *Stercoris* and *Bacillus altitudinis*) could improve the maturity and fertility of microorganisms, prolong the thermophilic stage, and increase the seed germination index (GI) [14]. The microbial inoculation (MI) mainly consisted of *Azotobacter chroococcum*, *Bacillus subtilis*, *Saccharomonospora* sp. and *Streptomyces albidoflavus*, which can enhance humification and microbial enzyme activity [16]. Duan et al. showed that inoculation with 2% *Bacillus subtilis* during the composting of cattle manure and wheat stalks accelerated the composting maturation and improved the seed GI [17]. In addition, microbial inoculants can accelerate the degradation of lignocellulose [18], promote compost maturation [19], and improve the efficiency of composting [20]. Moreover, Bacillus has been widely used in composting, but its application at low temperatures has rarely been studied.

Mushroom residue, which is a by-product of edible fungi production [21], is abundant in protein, polysaccharides, mycelium, carbohydrate, and mineral elements such as nitrogen, phosphorus, and potassium. It is a valuable renewable agricultural resource and is a kind of agricultural waste [22–24]. According to the statistics of the China Edible Fungus Association, the total output of edible fungi in China in 2020 was 40.6143 million tons, with a total output of 346.565 billion (https://www.cefa.org.cn/web/index.html, accessed on 16 August 2022). For every 200 g of mushrooms produced, 800 g of mushroom residue is discarded as garbage [25], so the recycling and utilization of mushroom residue would reduce resource waste and environmental pollution to a certain extent [24]. Mushroom residue can be replanted mushrooms [22] and or be used as compost for field organic fertilizer development [23]. As improper disposal of mushroom residue and livestock manure (the utilization rate of livestock manure is <60% [26], despite the rapid increase in large- and medium-sized livestock) can cause serious environmental pollution, it is necessary to find effective ways to increase the utilization rate as soon as possible.

Low winter temperatures or low mean annual temperatures in many areas present challenges to composting. For example, Xining, Qinghai Province, which is on a continental plateau, has an average annual temperature of 7.6 °C and a semi-arid climate. Researchers recommend inoculating compost with cold-adapted, highly active microorganisms to speed up OM decomposition and ensure the normal start-up of compost [27]. In this study, mushroom residue and sheep manure were used as raw materials, and composite microbial agent (CMA) were added to conduct a composting experiment in Xining city, Qinghai province. The overall objectives of this study were to (a) identify a mixed-strain CMA with high cellulase activity and adaptation to low temperatures; (b) explore the effect of the CMA on the changes in physicochemical parameters during composting; and (c) study the effect of the CMA on the changes in the microbial community and the relationship between the microbial community structure and physicochemical parameters.

## 2. Materials and Methods

### 2.1. Strain Screening and Physiological and Biochemical Characteristics

2.1.1. Strain Isolation and Growth Characteristics

First, 5 g of cottonseed shell samples (Dabaozi Qinghai Huitian Agricultural Planting Base, Xining, Qinghai Province) were weighed and added to a triangular bottle containing 45 mL sterile water, and then shaken for 10 min before gradient dilution. Plate coating was used for the cultivation of lactic acid bacteria (LAB), bacillus, aerobic bacteria, and coliform bacteria. After purification of the various strains twice, single colonies were selected and cultured on oyster mushroom powder medium (2% oyster mushroom powder + 2% agar) and mushroom residue powder medium (2% mushroom residue powder + 2% agar) at 37 °C for 24 h, and strains with good growth were selected for follow-up experiments. The temperature tolerance was assessed at 10 °C and 15 °C in nutrient agar (NA) medium (0.5% beef extract, 1.0% peptone, 0.5% NaCl, and 1.7% agar at pH 7.0) under aerobic conditions for 24 h. The strains that grew better at low temperatures (10–15 °C) were selected for the next step, and the strain that grew best at low temperatures (10–15 °C) was selected for identification by sequencing.

2.1.2. Congo Red Assay to Assess Cellulase-Producing Activity

The purified strains were inoculated on carboxymethyl cellulose sodium (CMC-Na) solid medium (1.0% yeast meal, 1.0% CMC-Na, 0.5% NaCl, 0.1% $KH_2PO_4$, 0.02% $MgSO_4$, and 1.8% agar at pH 7.0). The plates were incubated at 37 °C for 24 h. After incubation, all plates were stained with Congo red solution for 30 min and then washed with 1 M NaCl solution for 5 min. The diameters of the hydrolytic zones (D) and colonies (H) were measured using a vernier caliper. The hydrolytic zone diameter/colony diameter ratio was then calculated to use as the criterion for selecting the best strain. The strain with the highest ratio was selected for identification by sequencing [28].

2.1.3. Antibacterial Activity

To conduct a mutual inhibition test on the two strains obtained in the prior steps, screening technique described by Wang et al. [29] was used with appropriate adjustments. First, 20 mL NA medium was poured into a plate and allowed to solidify. A strain (optical density at 600 nm [$OD_{600 nm}$] = 1) was transferred into 5 mL NA agar medium (<50 °C), with a 3% inoculation rate. It was then added to the upper layer of the 20 mL NA medium. After solidification, four holes were made on the surface of the NA solid medium with a perforator. Next, 200 μL cell-free supernatant of another strain cultured overnight was added to the holes. The diameter of the inhibition zone was determined after 24 h of incubation at 37 °C.

### 2.2. Composting Process and Sample Collection

2.2.1. Raw Material and Preparation of Inoculants

Fresh sheep manure and mushroom residue were used as the composting raw materials. The basic physical–chemical properties of raw materials are shown in Table 1. Fresh sheep manure was obtained from Daotang River town in Hainan Tibetan Autonomous Prefecture of Qinghai province, China, and the mushroom residue was collected from Huitian ecological park in Xining city, Qinghai province, China. The raw materials were pulverized to 80-mesh by a grinder after collection. The composite microbial agent (CMA) was prepared at the Key Laboratory of Ion Beam Bioengineering, Zhengzhou University. The CMA consisted of *Bacillus* (*B.*) *cereus* QS7 and *B. pumilus* QM6.

**Table 1.** The basic physicochemical properties for composting raw materials.

| Raw Materials | OM (%) | TOC (%) | TN (%) | C/N Ratio | TP (%) | TK (%) | pH | EC (mS/cm) |
|---|---|---|---|---|---|---|---|---|
| mushroom residue | 56.03 | 44.57 | 1.31 | 33.98 | 0.27 | 1.08 | 6.19 | 2.75 |
| sheep dung | 34.11 | 24.3 | 1.34 | 18.15 | 0.38 | 1.4 | 8.71 | 2.95 |

Note: OM, organic matter; TOC, total organic carbon; TN, total nitrogen; C/N, TOC/TN; TP, total phosphorus; TK, total potassium; EC, electrical conductivity.

### 2.2.2. Experimental Design and Sampling

The experiment was conducted at Qinghai Special Fertilizer Co., Ltd. (Xining, Qinghai, China), from July to September 2021. Four groups, designated B1, B2, T1, and T2, were used in the composting experiments. Mushroom residue and fresh sheep manure were mixed at a ratio of 5:1 (total 14.2 tons) or 9:1 (total 13.4 tons) by volume, and correspond to B and T treatments, respectively. B1, and T1 were the experimental groups inoculated with 0.1% CMA ($1 \times 10^9$ colony-forming units/mL; inoculant volume/wet weight of composting sample), and and B2, T2 were the control groups without inoculation.

The size of each compost pile was as follows: 1.35 m height × 4 m width × 7 m top length × 8 m bottom length (Figure 1). The moisture content of each compost pile was adjusted to 50−60% using tap water. The composting process lasted 60 days. Samples from the original compost (day 0) were taken, along with samples on days 5, 10, 15, 20, 30, 45, and 60, after stirring the composting substrate using a machine. The samples were obtained from the middle layer of each compost pile using a five-point sampling method, the subsamples were thoroughly mixed for subsequent analysis and then stored at −20 °C. Each sample was divided into two parts for physicochemical analysis and sequencing analysis.

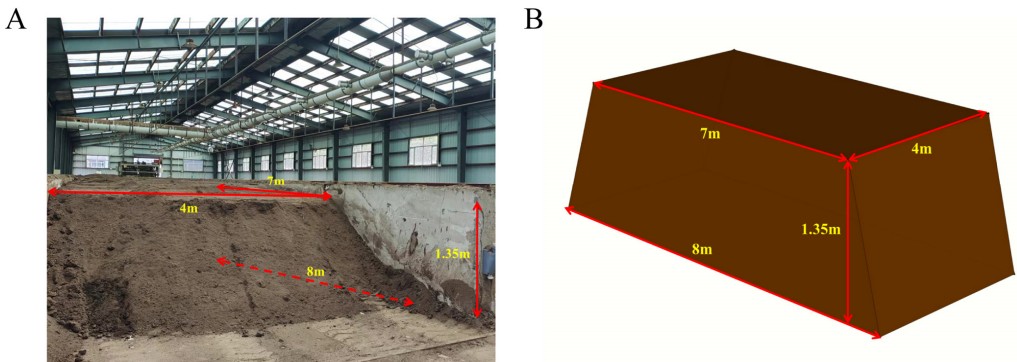

**Figure 1.** Compost distribution map. (**A**) compost site photo, (**B**) compost model.

### 2.3. Determination of Physicochemical Properties of Compost Samples

The ambient temperature and the compost temperature (20 cm depth from the compost surface) were recorded at 12:00 every day. The well-mixed compost samples were weighed, dried at 50 °C, and reweighed. The ratio of the amount of water lost to the wet weight of the sample was defined as the moisture content [30]. The pH and electrical conductivity (EC) values were determined by a pH meter (PHS-3C, Shanghai Yueping Scientific Instrument Manufacturing Co., Ltd., Suzhou, China) and a conductivity meter after the sample was mixed with distilled water at a ratio of 1:9.

Additionally, rapeseeds were cultured in the compost samples, with a 1:9 ratio of sample to distilled water. Distilled water was used as the control group. After 72 h, the germination rate and root length of the rapeseeds were measured and the germination index (GI) was calculated using Equation (1) [14].

Furthermore, 2 g of fresh compost sample was weighed and placed in a crucible with $w_0$ weight. The total weight was recorded as $w_1$. The samples in the crucible were dried at 105 °C for 24 h in an oven, weighed, recorded as $w_2$, and then the crucibles and the dried samples were dried at 550 °C for 6 h in a muffle furnace. The samples were cooled, weighed again, and recorded as $w_3$. The total carbon content (TOC) was calculated according to Equation (2). Organic matter content (OM), total nitrogen content (TN), total phosphorus content (TP), and total potassium content (TK) were determined based on descriptions in Agricultural Industry Standard NY 525-2021 for organic fertilizers. Total porosity, aeration porosity (AP), and water-holding porosity (WHP) were measured according to the method described by Wang et al. [30]. The ratio of compost weight to initial compost volume indicated bulk density.

$$\text{GI (\%)} = (\text{seed germination rate of treatment} \times \text{root length})/(\text{seed germination rate of control}) \times \text{root length} \times 100 \quad (1)$$

$$\text{TOC (\%)} = (w_2 - w_3)/(w_2 - w_0)/1.8 \times 100\% \quad (2)$$

### 2.4. Microbial Community Analysis

#### 2.4.1. DNA Extraction

High-throughput sequencing techniques were used to analyze microbial communities in samples during composting, including bacteria and fungi. The FastDNA® Spin Kit for Soil (MP Biomedicals, Solon, OH, USA) was used to extract genomic DNA from the compost samples according to the manufacturer's instructions. The DNA extracts were checked for quality using 1% agarose gel electrophoresis, and the concentration and purity of DNA extracts were determined using a spectrophotometer (NanoDrop 2000, Thermo Scientific, Wilmington, DE, USA).

#### 2.4.2. PCR Amplification

The obtained DNA was used for polymerase chain reaction (PCR) amplification by an ABI GeneAmp® 9700 PCR thermocycler (ABI, Oakland, CA, USA). The V3-V4 region of the bacterial 16S rRNA gene was amplified with primer pairs 338F (5′-ACTCCTACGGGAGGC AGCAG-3′) and 806R (5′-GGACTACHVGGGTWTCTAAT-3′), while the primers ITS1F (5′-CTTGGTCATTTAGAGGAAGTAA-3′) and ITS2R (5′-GCTGCGTTCTTCATCGATGC-3′) was used to amplify the ITS1 region of fungal ITS gene. The PCR amplification was performed as follows: (i) denaturation at 95 °C for 3 min, (ii) 27 cycles (for bacteria) and 35 cycles (for fungi) of denaturing at 95 °C for 30 s, annealing at 55 °C for 30 s and extension at 72 °C for 45 s (iii) followed by a single extension at 72 °C for 10 min, and end at 10 °C. The PCR reaction was performed in triplicate for each sample. Sequencing was performed using the Illumina Miseq PE300 platform by Majorbio Bio-Pharm Technology Co., Ltd. (Shanghai, China). The data were analyzed on the online platform of Majorbio Cloud Platform (http://www.majorbio.com/, accessed on 11 July 2022).

### 2.5. Statistical Analysis

All chemical and physical analyses were performed in triplicate. Statistical analysis was conducted using SPSS 22.0, and statistical graphs were performed using Origin 2017 software. The significant difference was determined at $p < 0.05$ using Duncan's multiple range method comparisons.

## 3. Results

### 3.1. Characteristics of the CMA

The morphological, physiological, and biochemical characteristics of the two selected strains are shown in Table 2. Strain QS7 grew very well on oyster mushroom powder medium and mushroom residue powder medium, while strain QM6 grew well on mushroom residue powder medium. Strain QS7 grew very well at 15 °C and well at 10 °C, and QM6 both grew at 10 °C and 15 °C. The D/H value of QS7 was 1.33 and that of QM6 was 7.41. QM6 had the highest cellulase-producing ability. Regarding the antibacterial activity of the two strains, QS7 and QM6 did not have a mutual inhibitory effect. They were mixed to produce the CMA for subsequent composting. QS7 was identified as *Bacillus* (*B.*) *cereus* and QM6 as *B. pumilus* by sequencing analysis.

**Table 2.** Morphological, physiological, and biochemical properties of the strains QS7 and QM6.

| Character | QS7 | QM6 |
|---|---|---|
| Shape | Rod | Rod |
| Gram stain | + | + |
| Growth on medium: | | |
| Oyster mushroom powder medium | +++ | − |
| mushroom residue powder medium | +++ | ++ |
| Growth at temp (°C): | | |
| 10 °C | ++ | + |
| 15 °C | +++ | + |
| Antibacterial activity: | | |
| QS7 | | − |
| QM6 | − | |

Note: For medium growth and temperature tolerance: +++, grow very well; ++, grow well; +, grow weakly; −, no growth. For antimicrobial activity, the inhibition zone contains the external diameter of the perforator (9.96 mm). Diameter of inhibition zone: −, less than 10 mm; +, 10–15 mm; ++, 15–20 mm; +++, 20–25 mm.

### 3.2. Changes in Temperature during Composting

The core temperatures of the various compost piles were measured and showed three composting stages: heating up (days 0–4), high temperature (days 4–35), and cooling down (days 35–60). The changes in the temperature are illustrated in Figure 2. Although the temperature of each group increased rapidly after composting, the compost pile in T1 heated up the fastest (from 34.2 °C to 47.3 °C). The temperature of all compost piles reached >50 °C on day 2 of composting, with the temperature in T1 being the highest (52.8 °C). After composting, the temperature peaked at 61.1 °C for B1, 60.3 °C for B2, 62.8 °C for T1 and 61.4 °C for T2. The days on which the compost temperature exceeded 55 °C for B1, B2, T1, and T2 were days 20, 19, 29, and 29, respectively, while the days on which the compost temperature exceeded 60 °C were days 2, 1, 11, and 5, respectively. On day 34, the temperature of each pile began to decrease gradually, with the temperature decreasing faster in B1 and T1.

### 3.3. Changes in Physicochemical Properties during Composting

Throughout composting, the OM content was higher in T groups (T1 and T2) than in B groups (B1 and B2) on the whole (Figure 3A). The OM content increased significantly on day 5 of composting, but there was no significant difference between T1 and T2 treatments ($p < 0.05$). After composting (on day 60), the OM content decreased significantly ($p < 0.05$), and it was significantly lower in group B1 than in B2 ($p < 0.05$). Compared to on day 0, the OM content on day 60 in T1 and T2 treatments declined by 12.22% and 8.49%, respectively, and the degradation rate of OM was higher in T1 than in the other groups.

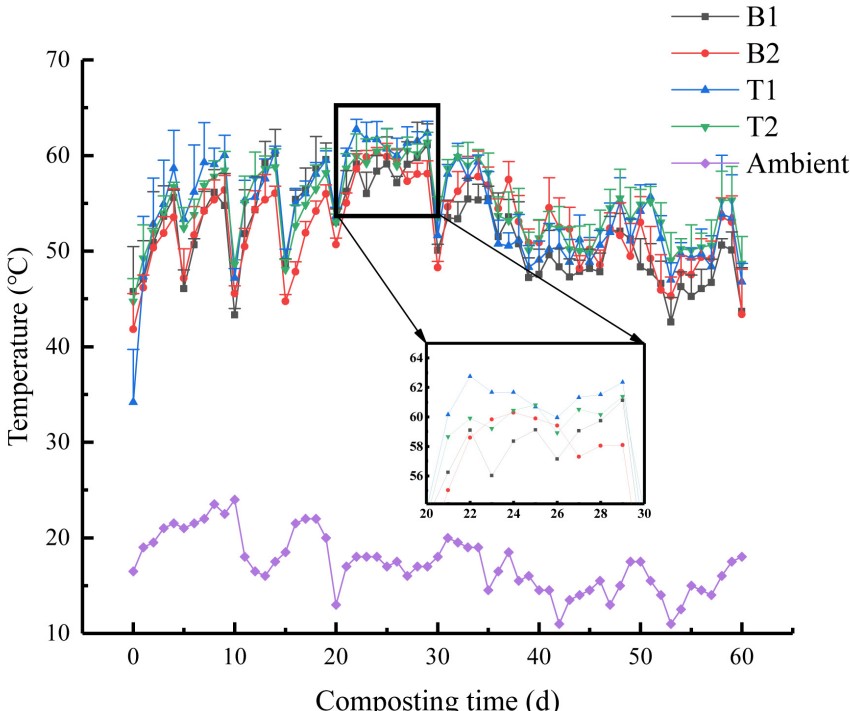

**Figure 2.** Evolution of temperature during composting. B1, $V_{mushroom\ residue}:V_{sheep\ manure}$ = 5:1, with CMA inoculation; B2, $V_{mushroom\ residue}:V_{sheep\ manure}$ = 5:1, without inoculation; T1, $V_{mushroom\ residue}:V_{sheep\ manure}$ = 9:1, with CMA inoculation; T2, $V_{mushroom\ residue}:V_{sheep\ manure}$ = 9:1, without inoculation.

Except for in the B2 treatment, TOC content first increased and then decreased (Figure 3C). The final TOC values were significantly lower ($p < 0.05$) than on day 0. Additionally, the reduction in TOC was greater in T1 than in T2. The TN content was higher in T1 and T2 groups than in other groups during the composting process, and the TN ratio of groups T1 and T2 increased to 1.94% and 1.82%, respectively (Figure 3D). The $NH_4^+$-N content in all groups increased to a high level, plateaued, decreased, and finally plateaued at a low level (Figure 3E). Both phosphorus and potassium were the main nutrients required for plant growth. The total phosphorus (TP) content fluctuated greatly (Figure 3F). After composting, the TP content was higher in T1 and T2 treatments than in the other treatments. The total potassium (TK) content increased and then gradually plateaued in the later composting period (Figure 3G). Regarding C/N, an important indicator of the maturity of compost [31], on day 10, C/N was higher in T1 than in the other groups. C/N was significantly lower at the end of composting than on day 0 ($p < 0.05$) (Figure 3B). Affected by the ratio of mushroom residue to manure, C/N was significantly lower in the B groups than in the T groups at the end of composting ($p < 0.05$). The final C/N ratios in B1, B2, T1 and T2 were 12.71%, 12.21%, 14.91% and 15.79%, respectively.

The bulk density, total porosity, aeration porosity (AP), water-holding porosity (WHP), and AP/WHP of the four compost piles during the composting were measured (Figure 3H–L). The bulk density in B1 and B2 groups on day 0 were 0.55 and 0.47 g/cm$^3$, respectively, which were higher than those of the other groups. This resulted in the bulk density being significantly higher in B1 and B2 than in the other groups at the end of composting. In addition, the AP value was higher in T1 than in the other groups.

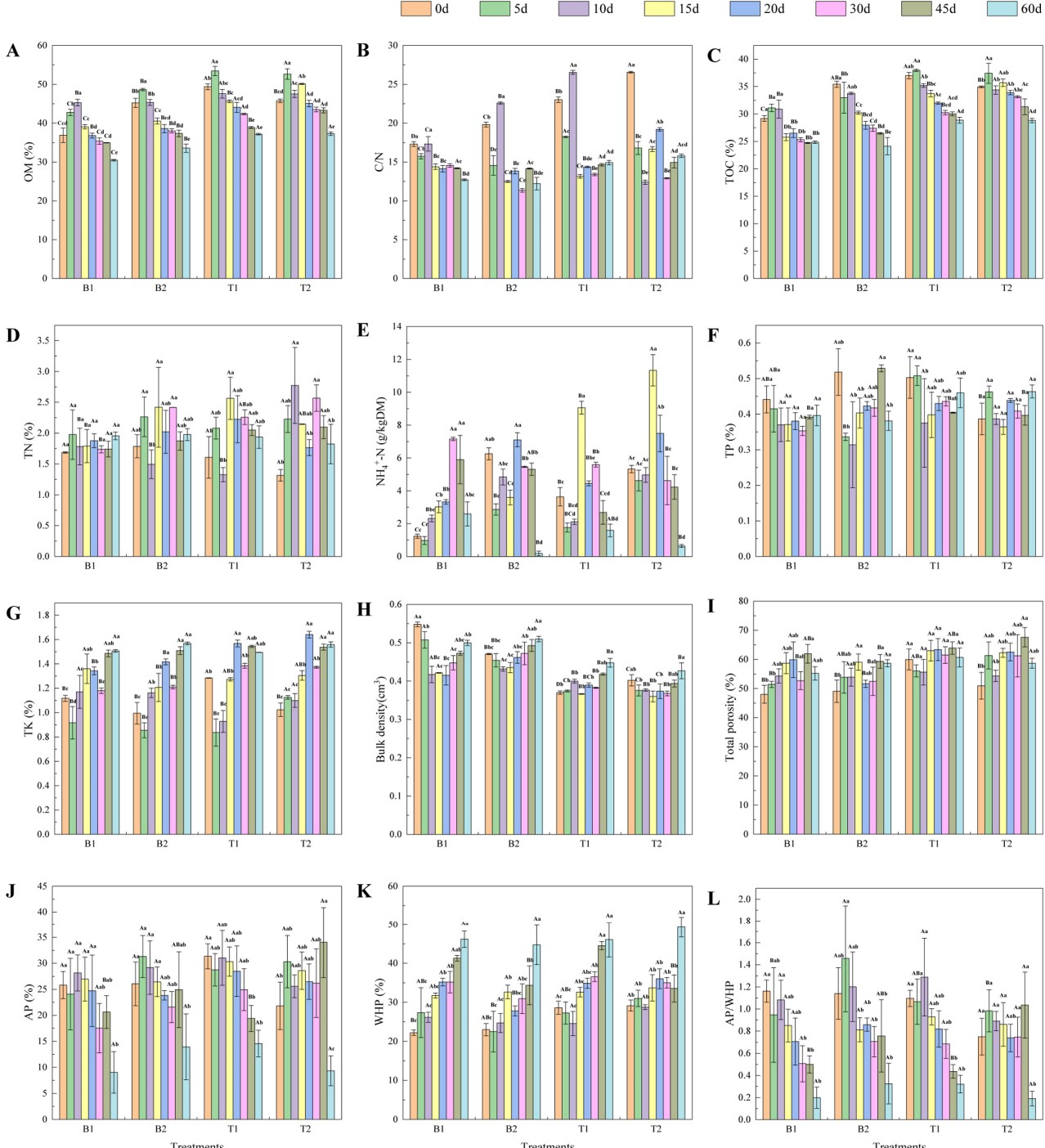

**Figure 3.** Variations of physicochemical parameters during composting. (**A**) organic matter (OM), (**B**) C: N, (**C**) Total organic carbon (TOC), (**D**) Total nitrogen (TN), (**E**) $NH_4^+$-N concentration, (**F**) Total phosphorus (TP), (**G**) Total potassium (TK), (**H**) bulk density, (**I**) total porosity, (**J**) aeration porosity (AP), (**K**) water-holding porosity (WHP), (**L**) AP/WHP. B1, $V_{mushroom\ residue}$:$V_{sheep\ manure}$ = 5:1, with CMA inoculation; B2, $V_{mushroom\ residue}$:$V_{sheep\ manure}$ = 5:1, without inoculation; T1, $V_{mushroom\ residue}$:$V_{sheep\ manure}$ = 9:1, with CMA inoculation; T2, $V_{mushroom\ residue}$:$V_{sheep\ manure}$ = 9:1, without inoculation. The significant difference within the same group (different days of the same treatment) is represented by lowercase letters, while the significant difference among the groups (different treatments on the same day) is represented by uppercase letters ($p < 0.05$).

The pH increased gradually after the beginning of composting (Table 3). Subsequently, the pH in B1, B2, T1, and T2 on day 60 of composting decreased to 8.51, 8.28, 8.35, and 8.52, respectively. In the end, the pH of all groups was weakly alkaline.

**Table 3.** Dynamic changes of pH and EC during composting.

| Items | Days (d) | Treatments | | | | SEM | *p* Value | | |
|---|---|---|---|---|---|---|---|---|---|
| | | **B1** | **B2** | **T1** | **T2** | | **T** | **D** | **T × D** |
| pH | 0 | $7.97 \pm 0.02$ Ac | $7.67 \pm 0.03$ Bd | $6.89 \pm 0.02$ Df | $7.09 \pm 0.03$ Cg | 0.041 | <0.05 | <0.05 | <0.05 |
| | 5 | $7.40 \pm 0.02$ Ad | $7.37 \pm 0.03$ Ae | $7.16 \pm 0.07$ Be | $7.26 \pm 0.02$ Bf | | | | |
| | 10 | $8.45 \pm 0.06$ Ab | $8.24 \pm 0.04$ Bc | $8.05 \pm 0.01$ Cd | $7.67 \pm 0.05$ De | | | | |
| | 15 | $8.50 \pm 0.04$ Bb | $8.59 \pm 0.03$ ABb | $8.68 \pm 0.10$ Ab | $8.16 \pm 0.05$ Cd | | | | |
| | 20 | $8.71 \pm 0.07$ Aa | $8.77 \pm 0.04$ Aa | $8.59 \pm 0.05$ Bb | $8.72 \pm 0.02$ Ab | | | | |
| | 30 | $8.81 \pm 0.02$ ABa | $8.84 \pm 0.01$ Aa | $8.69 \pm 0.02$ Bb | $8.72 \pm 0.02$ Bb | | | | |
| | 45 | $8.78 \pm 0.07$ Ba | $8.80 \pm 0.05$ ABa | $8.83 \pm 0.05$ ABa | $8.90 \pm 0.01$ Aa | | | | |
| | 60 | $8.51 \pm 0.07$ Ab | $8.28 \pm 0.11$ Bc | $8.35 \pm 0.10$ Bc | $8.52 \pm 0.06$ Ac | | | | |
| EC (mS/cm) | 0 | $1.43 \pm 0.02$ Cd | $1.68 \pm 0.05$ Bc | $2.45 \pm 0.07$ Aa | $1.77 \pm 0.08$ Bd | 0.066 | <0.05 | <0.05 | <0.05 |
| | 5 | $1.08 \pm 0.02$ Ce | $1.36 \pm 0.01$ Bd | $1.13 \pm 0.02$ Cd | $2.06 \pm 0.06$ Abc | | | | |
| | 10 | $1.56 \pm 0.01$ BCd | $1.72 \pm 0.14$ Bc | $1.52 \pm 0.01$ Cc | $2.18 \pm 0.17$ Abc | | | | |
| | 15 | $1.97 \pm 0.17$ ABc | $1.84 \pm 0.06$ Bbc | $2.02 \pm 0.06$ Ab | $2.06 \pm 0.01$ Abc | | | | |
| | 20 | $2.23 \pm 0.06$ Ab | $1.85 \pm 0.05$ Bbc | $1.88 \pm 0.09$ Bb | $1.88 \pm 0.08$ Bcd | | | | |
| | 30 | $2.44 \pm 0.02$ Ab | $2.02 \pm 0.03$ Bb | $2.41 \pm 0.01$ Aa | $2.00 \pm 0.07$ Bc | | | | |
| | 45 | $2.36 \pm 0.11$ ABb | $2.00 \pm 0.02$ Cb | $2.46 \pm 0.05$ Aa | $2.25 \pm 0.01$ Bb | | | | |
| | 60 | $2.91 \pm 0.09$ Aa | $2.59 \pm 0.22$ Ba | $2.09 \pm 0.07$ Cb | $2.47 \pm 0.05$ Ba | | | | |

Data are shown as the average values $\pm$ standard deviation (biological replicates, n = 3). B1, $V_{\text{mushroom residue}}$:$V_{\text{sheep manure}}$ = 5:1, with CMA inoculation; B2, $V_{\text{mushroom residue}}$:$V_{\text{sheep manure}}$ = 5:1, without inoculation; T1, $V_{\text{mushroom residue}}$:$V_{\text{sheep manure}}$ = 9:1, with CMA inoculation; T2, $V_{\text{mushroom residue}}$:$V_{\text{sheep manure}}$ = 9:1, without inoculation; T, treatments; D, days. Lowercase letters indicate vertical significance (different days for the same treatment); uppercase letters indicate horizontal significance (different treatments on the same day).

EC and GI are usually used to reflect the maturity of the finished compost [32]. EC, which reflects the concentration of water-soluble salt in the compost, is the concentration of soluble ions in the compost products [33]. High EC values can affect seed germination and reduce plant growth rate [34], indicating that compost products may not be suitable for direct use as plant growth substrate, otherwise it will affect the normal growth of plants. GI judges the maturity of compost products by evaluating phytotoxicity [35]. EC increased first and then decreased; the final EC values in B1, B2, T1, and T2 groups were 2.91, 2.59, 2.09, and 2.47 mS/cm, respectively, being significantly lower in T1 than other groups ($p < 0.05$). Table 4 shows the germination rate and GI obtained by planting rape seeds with the compost end products. There was no significant difference in the germination rate among the four groups, while GI was significantly higher in B1, B2, and T1 than in T2 ($p < 0.05$). In addition, the germination rate and GI were the highest in the T1 group among the four groups.

**Table 4.** Germination rate and germination index of rape seeds.

| Treatment | B1 | B2 | T1 | T2 |
|---|---|---|---|---|
| Germination rate (%) | $93.33 \pm 5.77$ a | $98.33 \pm 2.89$ a | $100.00 \pm 0.00$ a | $96.67 \pm 5.77$ a |
| Germination index (%) | $99.72 \pm 17.82$ a | $90.15 \pm 14.51$ a | $106.86 \pm 6.86$ a | $83.06 \pm 5.49$ b |

Note: B1, $V_{\text{mushroom residue}}$:$V_{\text{sheep manure}}$ = 5:1, with CMA inoculation; B2, $V_{\text{mushroom residue}}$:$V_{\text{sheep manure}}$ = 5:1, without inoculation; T1, $V_{\text{mushroom residue}}$:$V_{\text{sheep manure}}$ = 9:1, with CMA inoculation; T2, $V_{\text{mushroom residue}}$:$V_{\text{sheep manure}}$ = 9:1, without inoculation. The significant difference among the groups is represented by lowercase letters ($p < 0.05$).

*3.4. Changes in Bacterial and Fungal Communities and Diversity*

3.4.1. Diversity and Richness of Bacterial and Fungal Communities

Diversity indices reflect the microbial community dynamics during composting. As shown in Table 5, the diversity of the microbial communities in the composting process was studied, including the diversity and richness of the microbial communities, represented by the Shannon and Chao1 indices, respectively. The average Good's coverage values of bacteria and fungi in all samples were >98%, which indicated that the sequencing results could represent the real situation regarding microorganisms in the samples. On day 15, the Chao1 index was higher in B1 and T1 than in the other groups. From day 30, the Shannon index was lower in B1 and T1 than in other groups. Bacterial sequencing results showed that the operational taxonomic units (OTUs) and Chao1 index of the four groups increased first and then decreased, and the Chao1 index was higher from day 15 than on day 0 d. Regarding the fungal sequencing results, the Shannon increased in T2, while it decreased at the early stage of composting in the other groups. Moreover, on day 60, the Chao1 index was higher in B1 and T1 (194.48 and 193.78, respectively) than in other groups, with the largest increase in T1.

**Table 5.** Richness and Diversity of Microbial Community.

| Days | Treatments | Bacteria | | | | Fungi | | | |
|---|---|---|---|---|---|---|---|---|---|
| | | OTUs | Shannon | Chao1 | Coverage | OTUs | Shannon | Chao1 | Coverage |
| 0 d | B1 | 944 | 4.66 | 938.84 | 0.9925 | 279 | 1.89 | 189.75 | 0.9994 |
| | B2 | 580 | 3.85 | 592.23 | 0.9954 | 154 | 1.50 | 138.88 | 0.9994 |
| | T1 | 436 | 2.87 | 450.25 | 0.9963 | 89 | 1.77 | 87.17 | 0.9998 |
| | T2 | 410 | 2.90 | 381.09 | 0.9972 | 169 | 1.53 | 123.78 | 0.9995 |
| 15 d | B1 | 1337 | 4.95 | 1389.47 | 0.9882 | 362 | 1.47 | 229.82 | 0.9995 |
| | B2 | 1324 | 4.51 | 1298.95 | 0.9907 | 326 | 1.05 | 205.01 | 0.9996 |
| | T1 | 1096 | 4.27 | 1043.57 | 0.9937 | 251 | 1.16 | 154.97 | 0.9998 |
| | T2 | 575 | 1.62 | 548.80 | 0.9963 | 204 | 2.28 | 129.03 | 0.9999 |
| 30 d | B1 | 1377 | 4.56 | 1356.70 | 0.9914 | 381 | 3.53 | 236.80 | 0.9998 |
| | B2 | 1514 | 4.96 | 1435.93 | 0.9905 | 369 | 2.81 | 219.25 | 0.9998 |
| | T1 | 1312 | 4.69 | 1260.05 | 0.9921 | 329 | 3.13 | 189.08 | 0.9998 |
| | T2 | 1129 | 4.69 | 1085.86 | 0.9937 | 282 | 2.95 | 174.85 | 0.9998 |
| 60 d | B1 | 991 | 3.42 | 1013.72 | 0.9920 | 315 | 2.73 | 194.48 | 0.9998 |
| | B2 | 1079 | 3.68 | 1012.50 | 0.9917 | 282 | 2.72 | 168.31 | 0.9998 |
| | T1 | 1215 | 4.20 | 1175.42 | 0.9907 | 318 | 2.03 | 193.78 | 0.9997 |
| | T2 | 1170 | 4.42 | 1138.24 | 0.9918 | 242 | 2.35 | 153.28 | 0.9998 |

Note: B1, $V_{mushroom\ residue}$:$V_{sheep\ manure}$ = 5:1, with CMA inoculation; B2, $V_{mushroom\ residue}$:$V_{sheep\ manure}$ = 5:1, without inoculation; T1, $V_{mushroom\ residue}$:$V_{sheep\ manure}$ = 9:1, with CMA inoculation; T2, $V_{mushroom\ residue}$:$V_{sheep\ manure}$ = 9:1, without inoculation.

3.4.2. Principal Component Analysis (PCA)

The PCA of the genus-level microbes in the four groups is shown in Figure 4. Regarding the bacterial community composition, T1 and T2 groups were significantly separated on days 15 and 30 ($p < 0.05$) (Figure 4B,C). On day 60, the difference in the bacterial communities was greater between T1 and T2 than between B1 and B2 (Figure 4D). Regarding the fungal community composition, B2 and T2 were similar on day 0 and were significantly separated from the other groups ($p < 0.05$) (Figure 4E). After composting, all groups overlapped with T1 (Figure 4F,G). As shown in Figure 4H, the sample ellipse areas of the four

groups were highly different, and the area of the group ellipse was larger in B1 and T1 than in B2 and T2.

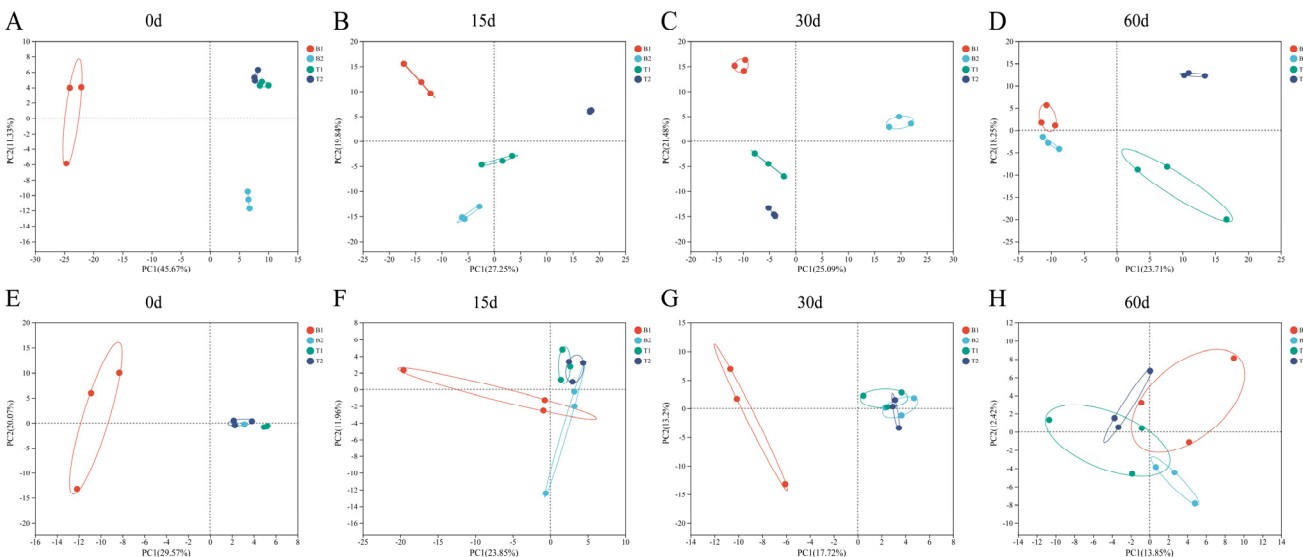

**Figure 4.** Principal component analysis (PCA) of bacterial and fungi community on the genus level during fermentation. (**A–D**): PCA of the bacterial community on 0 d, 15 d, 30 d, and 60 d, respectively; (**E–H**): PCA of fungi community on 0 d, 15 d, 30 d, and 60 d, respectively. B1, $V_{\text{mushroom residue}}$:$V_{\text{sheep manure}}$ = 5:1, with CMA inoculation; B2, $V_{\text{mushroom residue}}$:$V_{\text{sheep manure}}$ = 5:1, without inoculation; T1, $V_{\text{mushroom residue}}$:$V_{\text{sheep manure}}$ = 9:1, with CMA inoculation; T2, $V_{\text{mushroom residue}}$:$V_{\text{sheep manure}}$ = 9:1, without inoculation.

### 3.4.3. Change in Bacterial and Fungal Communities
Bacterial Composition Analysis

The microbial community composition during composting is shown in Figure 5A–D, regarding both bacterial and fungal communities. Figure 5A shows the changes in relative abundances of bacteria at the phylum level. The top five predominant bacterial phyla were Firmicutes, Actinobacteria, Bacteroidetes, Chloroflexi, and Proteobacteria, accounting for 93.46–99.84% of the total bacteria. In the early stage of composting, Firmicutes was much higher in T groups (T1 and T2) than the B groups (B1 and B2), while the relative abundance of Firmicutes and Bacteroidetes decreased and the relative abundance of Chloroflexi increased on day 60. The relative abundance of Proteobacteria increased over time in B2 and T2, decreased over time in B1, and was very low on day 0 in T1, and decreased further from day 15. Chloroflexi increased sharply with the end of the composting process, being 50.50%, 38.88%, 44.45%, and 33.36% in B1, B2, T1, and T2, respectively.

At the genus level (Figure 5C), the dominant bacteria in the mushroom residue were *Staphylococcus* and *Streptomyces*. During the composting process, the relative abundance of *Kurthia* decreased (<2% on day 30 and <1% on day 60 in all groups). The relative abundance of *Kurthia* in the T2 group was significantly higher than in other groups on day 15 ($p < 0.05$), and the relative abundance in the T1 group was higher than that in other treatments on day 60. The relative abundance of *Flavobacterium* in T1 was 30.62% on day 15, which was higher than that in other groups. The relative abundance of *Sphaerobacter* increased over time, being 21.77%, 5.78%, 6.91%, and 8.88% in B1, B2, T1, and T2, respectively. *Sphingobacterium* and *Corynebacterium* decreased and were extremely low on day 60 compared to day 0. The relative abundance of *Bacillus* in the T1 group was 0.014% on day 60, lower than in the other groups.

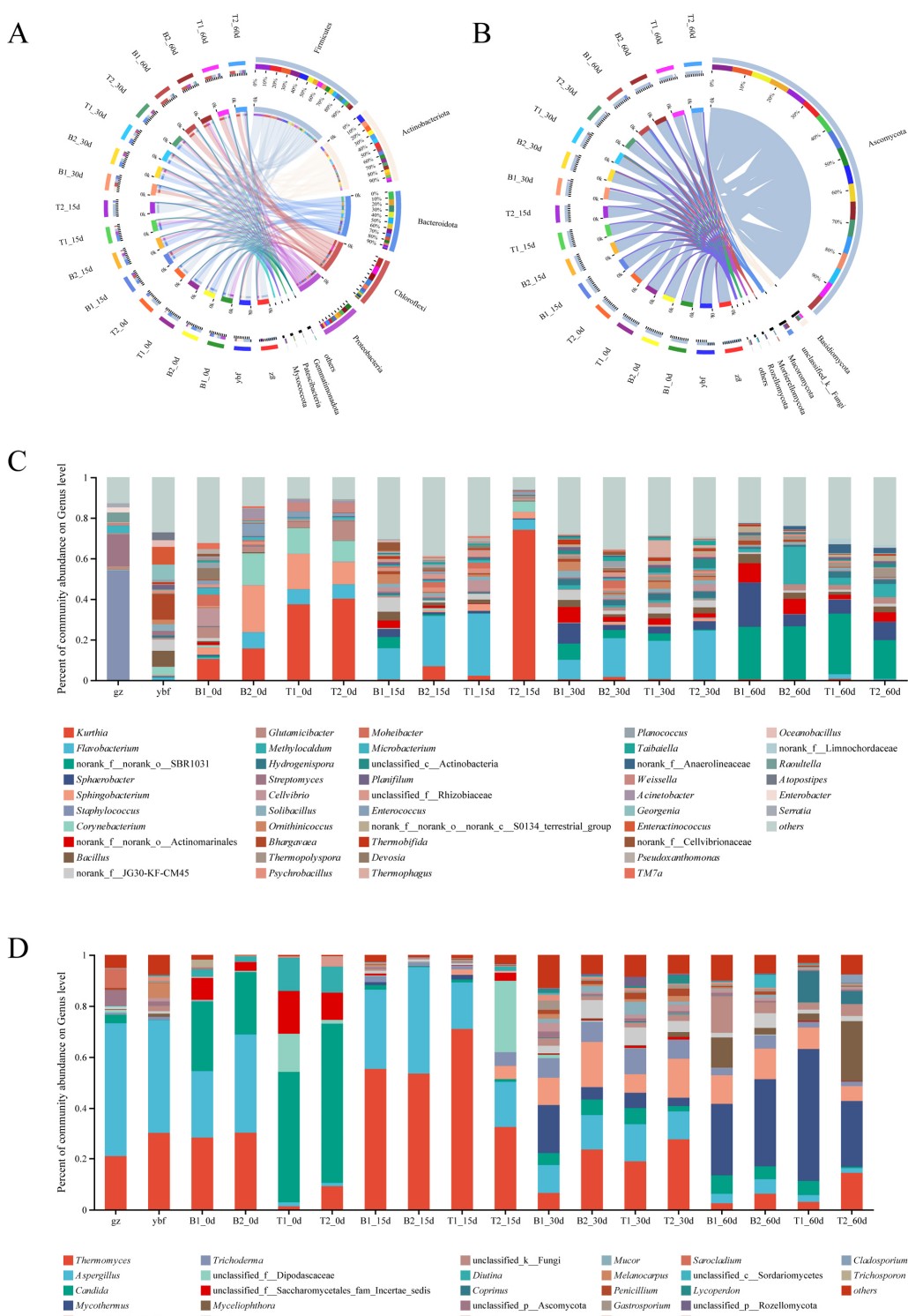

**Figure 5.** The relative abundance of bacteria at the phylum level (**A**), at the genus level (**B**), and fungi at the phylum level (**C**), at the genus level (**D**) among all samples. Taxa with abundance < 0.01% at the phylum level are classified as "other", while taxa with abundance < 0.02% at the genus level are classified as "other". B1, $V_{mushroom\ residue}$:$V_{sheep\ manure}$ = 5:1, with CMA inoculation; B2, $V_{mushroom\ residue}$:$V_{sheep\ manure}$ = 5:1, without inoculation; T1, $V_{mushroom\ residue}$:$V_{sheep\ manure}$ = 9:1, with CMA inoculation; T2, $V_{mushroom\ residue}$:$V_{sheep\ manure}$ = 9:1, without inoculation.

Fungal Composition Analysis

As for fungi, at the phylum level (Figure 5B), Ascomycota and Basidiomycota were the two major fungal phyla. The relative abundance of Ascomycota was >50% in all groups at all periods. The relative abundance of Basidiomycota increased at the later stage of composting. The changes in fungi at the genus level are shown in Figure 5D. The relative abundance of *Thermomyces* gradually increased and peaked on day 15 (high-temperature phase), topping out at 70.99% in T1. In addition, the relative abundance of *Candida* decreased. In contrast, the relative abundance of *Mycothermus* increased, being higher in B1 and T1 than B2 and T2 on day 30 and highest in the T1 group at the end of composting. At the end of composting, the relative abundance of *Coprinus* was the highest (12.30%) in the T1 treatment.

### 3.4.4. Correlation Analyses of Physicochemical Parameters with Bacterial and Fungal Genera

The heatmap in Figure 6 shows the Spearman correlations between the physicochemical properties of the compost samples and the microbial genera. *Flavobacterium* was significantly and positively correlated with pH ($r = 0.89$, $p \leq 0.001$) on day 15, but significantly and negatively correlated with C/N ($r = -0.73$, $p \leq 0.01$) on day 15 (Figure 6B). *Bacillus* was significantly and positively correlated with EC ($r = 0.81$, $p \leq 0.01$) on day 60, but significantly and negatively correlated with OM ($r = -0.78$, $p \leq 0.01$) and TOC ($r = -0.62$, $p \leq 0.05$) on day 60 (Figure 6D). Additionally, *Coprinus* was significantly and negatively correlated with EC ($r = -0.70$, $p \leq 0.05$) on day 60 (Figure 6H).

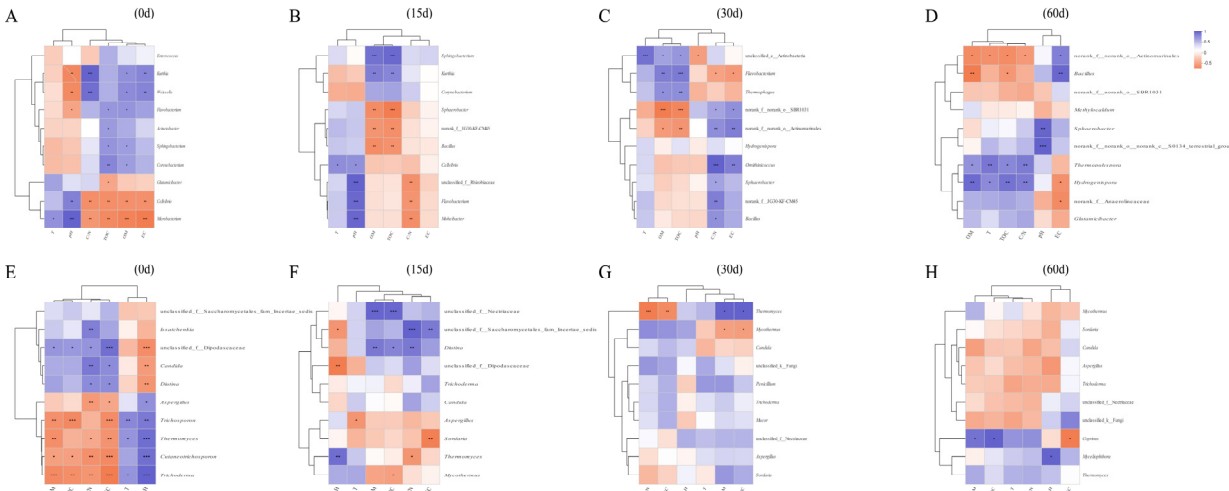

**Figure 6.** Spearman's correlation heatmaps of the abundance of the top 10 enriched bacteria and fungi at the genus level with physical and chemical properties during the composting process. Bacteria: 0 d (**A**), 15 d (**B**), 30 d (**C**), and 60 d (**D**) of composting. Fungi: 0 d (**E**), 15 d (**F**), 30 d (**G**), and 60 d (**H**) of composting. The X and Y axes are environmental factors and genus, respectively. Blue indicates a positive correlation, whereas orange indicates a negative correlation. B1, $V_{\text{mushroom residue}}$:$V_{\text{sheep manure}}$ = 5:1, with CMA inoculation; B2, $V_{\text{mushroom residue}}$:$V_{\text{sheep manure}}$ = 5:1, without inoculation; T1, $V_{\text{mushroom residue}}$:$V_{\text{sheep manure}}$ = 9:1, with CMA inoculation; T2, $V_{\text{mushroom residue}}$:$V_{\text{sheep manure}}$ = 9:1, without inoculation. *, $p \leq 0.05$; **, $p \leq 0.01$; ***, $p \leq 0.001$.

## 4. Discussion

Microbial inoculants have been widely used in composting systems, as they can accelerate OM degradation and phosphorus stabilization in the process of biological waste composting [36]. Cold-adapted microbial consortium can effectively promote the rise in the compost heap temperature to make it pass the start-up period [12]. In this study, suitable

strains were identified and then applied to compost comprising mushroom residue and sheep manure.

Microbial metabolic reactions and the degradation of OM all affect the change in compost temperature [11,34]. Changes in temperature during composting reflect microbial activity and composting efficiency, which is an important indicator affecting the microbial community [12,15,37]. Therefore, temperature distribution can also be used to measure microbial activity [5]. It has been reported that a temperature >55 °C lasts for >10 days during composting, meeting hygiene requirements by allowing the degradation of OM and the elimination of pathogenic bacteria and harmful pollutants [38,39]. In this study, in the heating and high-temperature stages, the heap temperature was slightly higher in B1 and T1 than in B2 and T2. A higher temperature promotes the maturation of compost [16]. Li et al. reported that adding exogenous microorganisms to the native microorganisms led to a synergistic effect on OM decomposition, similar to our study [19]. Additionally, in our study, *B. cereus* QS7 and *B. pumilus* QM6 have been shown to prolong the thermophilic period of compost [40]. At the later stage of composting, the temperature decreased due to the consumption of biodegradable substances [41]. Consistent with the results of He et al. [20], the temperature decreased faster in B1 and T1 than in the other groups. This may have occurred because the addition of CMA not only prolongs the thermophilic period, but also shortens the composting cycle and improves the composting efficiency.

Due to the change in the temperature and the conversion of OM, the physicochemical parameters during composting will change correspondingly. Our findings showed that under the same raw material ratio (ratio of mushroom residue to sheep manure), CMA inoculation increased the degradation of OM, which might be caused by the symbiosis between the native microorganisms and the CMA [42]. CMA increases microbial activity [43], allowing biodegradable OM to be degraded into nutrients suitable for microbial growth, thus initially increasing the composting temperature [44].

Compost is a metabolic process in which microorganisms participate, and TOC provides an organic carbon source for microbial growth [45]. The trend in TOC was consistent with the results of Meng et al. [46]. First, the increase in composting temperature accelerated cellulose degradation, but the enhanced activity of thermophilic microorganisms promoted the degradation of organic carbon into carbon dioxide, thus promoting the degradation of a large amount of TOC [13]. Additionally, the reduction in TOC was higher in the T1 group than T2 (under the same raw material ratio), which may be because the CMA promoted TOC degradation. The TOC values of the composts were all <40% on day 60, indicating that the composting was complete [31]. The increase in TN in all four groups was the result of the degradation of nitrogenous OM caused by the increase in temperature, which was similar to the initial increase in pH. Later, the $NO_x$ production via denitrification may explain the subsequent reduction in TN [47]. Biodegradation of nitrogen-containing organic compounds led to an initial rapid increase in $NH_4^+$-N, but the volatilization of ammonia and biological nitrification led to a subsequent decrease, keeping the $NH_4^+$-N content of the final products low [32,48]. The contents of TP and TK of compost were higher when the ratio of mushroom residue to sheep manure was high (9:1), and the nutrient content of compost products can be improved. In general, the findings revealed that CMA can accelerate the degradation of TOC and increase the accumulation of nitrogen. C/N is an important index to evaluate the maturity and stability of compost. The decrease in C/N in all four groups by the end of composting may essentially be related to the degradation of OM and the mineralization of nitrogen mentioned above. The C/N of the four groups was close to 15 on day 60, which met the maturity standard [14]. However, due to the difference in the initial C/N, the T value (the ratio (final C/N)/ (initial C/N)) of each group was calculated, and the results showed that they were all in line with the threshold of <0.75 mentioned by Itävaara et al. [49].

The pH value reflects microbial activity in the composting process. Too high or too low will affect the normal composting reaction. The pH value of 6.0–9.0 was the most suitable [44]. Consistent with the report of Wang et al. [23], the decomposition of

nitrogenous organic compounds led to the rapid volatilization of ammonia, which raised the pH. The subsequent decrease in pH may be related to the production of organic acids and carbon dioxide emissions [31,47]. On day 60, all groups reached the ripening standard (pH < 9) mentioned by Gou et al. [50]. In the case of a high mushroom residue ratio, inoculation of CMA can significantly reduce pH value. The initial decrease in EC may be due to the volatilization of ammonia and precipitation of mineral salts [46], followed by an increase in EC due to rapid OM degradation, which produced a large number of ions. In general, EC > 2.00 mS/cm has adverse effects on seed germination and plant growth [51]. After the end of composting, only the EC in the T1 group was close to 2.00 mS/cm, i.e., 2.09 mS/cm. EC was significantly lower in the T1 group than in the other groups. In addition, the EC value was lower under the condition of a high proportion of mushroom residue. As high EC will affect seed germination and reduce the plant growth rate [34], the compost type produced in our study may not be suitable for direct use as a plant growth substrate. In conclusion, the T1 group was more mature and more suitable as a plant culture substrate, which may be because CMA of raw materials with a high ratio of mushroom residue to sheep manure can promote the utilization of nutrient ions [9]. All samples reached the maturity index threshold (GI > 80%) [15]. For the same raw material ratio, GI was higher in the CMA groups than in the control groups, which may be related to the higher content of salt and heavy metals in the control groups [51]. Thus, adding CMA promoted seed germination and rapid growth of rapeseed.

CMA inoculation promoted the rapid increase in compost temperature and the growth of the microbial community during the thermophilic period, while the abundance of thermophilic bacteria eventually decreased the nutrient content and the community diversity. This explains why the Shannon index was higher in the thermophilic period. In all groups, the trends in the diversity indices were consistent with previous studies [43,52], which may be related to the increase in high-temperature resistant strains.

Regarding the bacteria, the dominant phyla in a composting study by Liu et al. [53] were similar to those in this study. A similar gradual decrease in the relative abundance of Firmicutes has been noted in studies of mixed compost involving cattle manure and wheat straw [54] and pig manure and corn straw [4]. Actinobacteriota is associated with lignocellulosic degradation and can adapt to harsh living conditions, such as high temperatures [39,55], which may explain why its relative abundance did not vary much during composting. On day 30, the relative abundance of Actinobacteriota in the CMA inoculated groups was higher than that in the uninoculated group at the same proportion, which may be related to the higher temperature in the CMA inoculated groups and may be the reason for the rapid degradation of OM. The relative abundance of Proteobacteria was higher in the CMA inoculation groups, which plays a key role in the degradation of small molecules and is closely associated with the mineralization of nitrogen-containing OM in the early composting period [56]. Chloroflexi is widely distributed, including autotrophic, heterotrophic, and mixed types [57–59], and its dramatic increase in all four groups may have been caused by the decrease in oxygen content in the compost due to the decrease in aeration porosity (Figure 3J). During composting, the basicity of the compost pile gradually increased, while the relative abundance of *Kurthia* decreased, indicating that *Kurthia* was not adapted for survival in an alkaline environment (the pH was about 8.5) and its relative abundance changes correspond to pH. *Flavobacterium*, as a member of Bacteroidota, was highest in T1 on day 15 and very low in all four groups after composting, probably due to its extreme dependence on oxygen, a property mentioned by Liu et al. [4]. It can promote the degradation of cellulose, hemicellulose, and lignin [14], which may be related to the rapid degradation of OM in the T1 group. The dominant bacterial genera in the compost of vegetable waste [60], dairy manure and wheat straws [61], sewage sludge [62], laying-hen manure [63], and biogas residue [20] in previous studies were different from those reported in this study, which may be the result of the combined effect of the compost materials, location, and environmental conditions.

Regarding the fungal phyla, Ascomycota and Basidiomycota, which were the two major fungal phyla in this study, both play a role in lignocellulose degradation [64] and have been reported to be efficient at utilizing nutrients in compost [65]. After composting, the total relative abundance of Ascomycota and Basidiomycota was the highest in the CMA-inoculated groups. Overall, the relative abundances of fungi at the genus and phylum levels in the four groups significantly changed over time. *Mycothermus* plays a key role in the degradation of cellulose and hemicellulose [66], which may account for the rapid degradation of OM in T1. The relative abundance of *Coprinus* increased in the late composting period, which was consistent with the results of the Klam and Bååth study [67]. The relative abundance of *Coprinus* was the highest in the T1 treatment, and *Coprinus* was significantly and negatively correlated with EC, which explained the reason for the lowest EC value in the T1 treatment. *Thermomyces*, *Aspergilus*, *Candida*, and *Mycothermus*, which were the dominant genera in this study, are thermophilic fungi that are common in compost. For example, in animal manure compost, the dominant fungal genera included *Aspergillus*, *Thermomyces*, and *Alternaria* [68], while *Candida*, *Aspergillus*, *Thermomyces*, and *Petriella* were the dominant fungi in municipal sludge compost [64]. In addition, compared to no inoculation, CMA inoculation increased the relative abundance of thermophilic fungi when the ratio of mushroom residue to sheep manure was 9:1, which was one of the reasons for the increase in compost temperature.

## 5. Conclusions

CMA inoculation (*Bacillus cereus* QS7 and *Bacillus pumilus* QM6) prolonged the thermophilic period of composting and shortened the composting cycle. In addition, CMA inoculation accelerated the degradation of OM and TOC. When the ratio of mushroom residue to sheep manure was 9:1, the nitrogen accumulation was increased, EC was significantly decreased and GI was increased. Moreover, the microbial community structure was affected by CMA inoculation. This study provides evidence to support the use of CMA inoculants in composting. Inoculation of a cold-adapted bacterial agent is a practical strategy for accelerating composting in low-temperature areas.

**Author Contributions:** M.W. (Min Wang): Methodology, Investigation, Formal analysis, Writing—original draft, Writing—review and editing. H.L.: Project administration, Writing—review and editing. L.X.: Investigation, Data curation. K.Z.: Investigation. Y.M.: Data curation. S.Z.: Investigation. M.W. (Ming Wu): Resources. Y.G.: Software. M.Z.: Investigation. H.P.: Writing—review and editing. Z.T.: Conceptualization, Supervision, Project administration, Funding acquisition. All authors have read and agreed to the published version of the manuscript.

**Funding:** This study was supported by the Scientific and Technological Cooperation Program for Assistance to Qinghai, grant number: 2021-QY-210; the third batch of "555 Talent Introduction and Gathering Project" flexible introduction of class II talents in Xining; the Key Scientific and Technological Project of Henan Province, grant number: 222102520027; and the Higher Education Institution Key Research Project Plan of Henan Province, grant number: 23A550004.

**Institutional Review Board Statement:** Not applicable.

**Informed Consent Statement:** Not applicable.

**Data Availability Statement:** The datasets presented in this study can be found in online repositories. The names of the repository/repositories and accession number(s) can be found below: https://www.ncbi.nlm.nih.gov/sra/PRJNA949816, accessed on 29 March 2023. The 16S rRNA gene sequence of *Bacillus* strains QM6, and QS7 used to support the findings of this study were deposited in the GenBank repository with accession numbers OQ703062, OQ703063, respectively. http://www.ncbi.nlm.nih.gov, accessed on 29 March 2023.

**Conflicts of Interest:** The authors declare no conflict of interest.

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
