# Peer review of "Screening of Cold-Adapted Strains and Its Effects on Physicochemical Properties and Microbiota Structure of Mushroom Residue Composting"

_fermentation, doi:10.3390/fermentation9040354_

Round 1

Reviewer 1 Report

-Page 1 Line 16  Abstract  abstract should be revised to add highlighted results with quantitative data, not just explaining in text for the highlighted results

-Page 3 Line 142 Table 1 why authors measure EC? what's your reason

-Page 5 Line 222  pleurotus ostreatus revise

-Page 5 Line 224  Both strains grew well at 10–15℃. please define author's criteria to "grow well"

-Page 6 Line 227 D/H value of QS7 was 1.33 and that of QM6 was 7.41. what about D/H of other strains? why authors selected these 2 strains from the collection

-Page 6 Line 235  Table 3. this table is not necessary

-Page 7 Line 251  Figure 2. it is not convincing that these samples have different temperature profiles as discussed in the above paragraph. The error bars are all overlapped.

-Page 7 Line 255  3.3. Changes in physicochemical properties during composting authors should add contents for discussion for each character to explain how inoculum has effect on each character and how B1 and T1 are different from each other and how they are different from control”

-Page 9 Line 306-308 . EC which reflects the salinity of the compost, how salinity related to the maturity of compost, please explain”

-Page 11 Line 358  3.4.3.1. Bacterial composition analysis   authors should add the function of each bacterial genus and explain the correlation of this function to the character of compost in fig 3e

-Page 13 Line 398  3.4.4. Correlation analyses of physicochemical parameters with bacterial and fungal genera authors should indicate what the level of r that is acceptable to consider as "correlate", like 0.62 is it considered as correlate or not correlate?

-Page 15 Line 495-516  low hemicellulose compounds in the solution most discussion here explain about the relationship between bacteria and cellulose degradation. However, this study does not have analysis of cellulose, hemicellulose and lignin contents?

Author Response

Dear Editors and Reviewers:

Thank you for inviting us to submit a revised draft of our manuscript entitled “Screening of cold-adapted strains and its effects on physicochemical properties and microbiota structure of mushroom residue composting”. We also appreciate the time and effort you and each of the reviewers have dedicated to providing insightful feedback on ways to strengthen our manuscript. Thus, it is with great pleasure that we resubmit our manuscript for further consideration. We have incorporated changes that reflect the detailed suggestions you have graciously provided. We also hope that our edits and the responses we provide below satisfactorily address all the issues and concerns you and the reviewers have noted. Revised portion are marked in red. The main corrections and responses are as follows:

Response to Reviewer 1 Comments

Point 1: Page 1 Line 16 Abstract “abstract should be revised to add highlighted results with quantitative data, not just explaining in text for the highlighted results

Response 1: Thank you for this suggestion. In the revision, we have changed the abstract content to “Composting is an effective way to dispose of agricultural waste; however, its application is limited in the winter and in areas with low average annual temperatures. This study screened out a composite microbial agent (CMA) including Bacillus (B.) cereus QS7 and B. pumilus QM6 that could grow at 10–15℃ and investigated the effects of the CMA as an inoculant on the physicochemical parameters and microbial communities of compost made from mushroom residue mixed with sheep manure. The results showed that CMA inoculation prolonged the days of high temperature above 60℃. When ratio of mushroom residue to sheep manure was at 9:1, compost increased the accumulation of nitrogen, and reduced the electrical conductivity (EC). Under this ratio, inoculation of CMA accelerated the degradation of organic matter (OM) (decreased by 12.22%) and total organic carbon (TOC) (decreased by 8.13%) and increased the germination rate and seed germination index (GI) to 100% and 106.86, respectively. Microbial community structure analysis showed that the relative abundance of Flavobacterium was 30.62% on day 15 was higher after CMA inoculation in the high mushroom residue condition, which was higher than that of other groups, and the relative abundance of thermophilic fungi increased. This study demonstrated that CMA improves the quality and efficiency of mushroom residue and sheep manure composting, and it provided evidence to improve the efficiency of low-temperature composting.”

Point 2: Page 3 Line 142 Table 1 “why authors measure EC? what's your reason

Response 2: Thank you for this suggestion. The reason for the determination of electrical conductivity (EC) has been mentioned in Lines 306-308 and Lines 478-480 of the original manuscript. We have detailed the reason in the revised version and have changed Lines 321-326 to “EC, which reflects the concentration of water-soluble salt of the compost, is the concentration of soluble ions in the compost products [33]. High EC values can affect seed germination and reduce plant growth rate [34], indicating that compost products may not be suitable for direct use as plant growth substrate, otherwise it will affect the normal growth of plants.” And we have changed Lines 506-507 to “In general, EC >2.00 mS/cm has adverse effects on seed germination and plant growth [51].”

  1. Tiquia, S.M. Reduction of compost phytotoxicity during the process of decomposition. 2010, 79(5), 506-512. doi: 10.1016/j.chemosphere.2010.02.040.
  2. Paula, F.S.; Tatti, E.; Abram, F.; Wilson, J.; O'Flaherty, V. Stabilisation of spent mushroom substrate for application as a plant growth-promoting organic amendment. Environ. Manage. 2017, 196, 476-486. doi: 10.1016/j.jenvman.2017.03.038.
  3. Ren, J.J.; Deng, L.J.; Li, C.Y.; Dong, L.P.; Li, Z.J.; Huhetaoli.; Zhang, J.; Niu, D.Z. Effects of added thermally treated penicillin fermentation residues on the quality and safety of composts. Environ. Manage. 2021, 283, 111984. doi: 10.1016/j.jenvman.2021.111984.

Point 3: Page 5 Line 222 pleurotus ostreatus “revise

Response 3: Thank you for this suggestion. We have changed “pleurotus ostreatus” to “oyster mushroom”. Please see Line 232. For other similar issues, we have carefully checked the manuscript and changed it in the revision. Please see Lines 115-116 and Table 2.

Point 4: Page 5 Line 224 Both strains grew well at 10–15℃. “please define author's criteria to "grow well"

Response 4: Thank you for this suggestion. We have amended this sentence to “Strain QS7 grew very well at 15℃ and well at 10℃, and QM6 both grew at 10℃ and 15℃.” Please see Lines 234-235. We have defined the growth conditions of the strain in different media and different temperatures in Table 2. Please refer to the sentence “For medium growth and temperature tolerance: +++, grow very well; ++, grow well; +, grow weakly; -, no growth.” in Lines 244-245.

Point 5: Page 6 Line 227 D/H value of QS7 was 1.33 and that of QM6 was 7.41. “what about D/H of other strains? why authors selected these 2 strains from the collection

Response 5: Thank you for this suggestion. The larger the ratio of hydrolytic zone diameter (D) to colony diameter (H), the higher the cellulase‑producing activity of the strain. QS7 grew well at 10-15℃, and QM6 had the highest cellulase-producing capacity, i.e., the ratio of hydrolytic zone diameter to colony diameter (D/H) was the highest, and QM6 also grew at 10-15℃. Therefore, these two strains were selected to make composite bacterial agent (CMA). Because of the large amount of data, only the data of the two strains finally selected are presented here.

Point 6: Page 6 Line 235 Table 3. “this table is not necessary

Response 6: Thank you for this suggestion. We have removed Table 3 and have changed the ordinals of the other tables in the revision.

Point 7: Page 7 Line 251 Figure 2. “it is not convincing that these samples have different temperature profiles as discussed in the above paragraph. The error bars are all overlapped.

Response 7: Composting is a complex system. In this study, the temperature was measured at the same time and at the same compost depth every day, and the average value of the measured results was used as the standard. According to the results of our present study, inoculation of CMA can prolong the high temperature days above 55℃ and 60℃, and the temperature decreased faster in the late composting period. In addition, when ratio of mushroom residue to sheep manure was at 9:1, inoculation of CMA accelerated the increase of compost temperature.

Point 8: Page 7 Line 255 3.3. Changes in physicochemical properties during composting “authors should add contents for discussion for each character to explain how inoculum has effect on each character and how B1 and T1 are different from each other and how they are different from control

Response 8: Thank you for this suggestion. We have added a description of each character and have already discussed the effects of CMA inoculation on different treatments in the revision.

Point 9: Page 9 Line 306-308. EC which reflects the salinity of the compost, “how salinity related to the maturity of compost, please explain

Response 9: Thank you for this suggestion. We have modified the sentence to “EC, which reflects the concentration of water-soluble salt of the compost, is the concentration of soluble ions in the compost products [33]. High EC values can affect seed germination and reduce plant growth rate [34], indicating that compost products may not be suitable for direct use as plant growth substrate, otherwise it will affect the normal growth of plants.” Please see Lines 321-326.

  1. Tiquia, S.M. Reduction of compost phytotoxicity during the process of decomposition. 2010, 79(5), 506-512. doi: 10.1016/j.chemosphere.2010.02.040.
  2. Paula, F.S.; Tatti, E.; Abram, F.; Wilson, J.; O'Flaherty, V. Stabilisation of spent mushroom substrate for application as a plant growth-promoting organic amendment. Environ. Manage. 2017, 196, 476-486. doi: 10.1016/j.jenvman.2017.03.038.

Point 10: Page 11 Line 358 3.4.3.1. Bacterial composition analysis “authors should add the function of each bacterial genus and explain the correlation of this function to the character of compost in fig 3e

Response 10: Thank you for this suggestion. We have added the function of the bacterial genus and have analyzed the relevance of this function to compost properties in the results and discussion section of the revision.

Point 11: Page 13 Line 398 3.4.4. Correlation analyses of physicochemical parameters with bacterial and fungal genera “authors should indicate what the level of r that is acceptable to consider as "correlate", like 0.62 is it considered as correlate or not correlate?

Response 11: Thank you for this suggestion. We have already mentioned in the revision in Lines 432-433 that “Blue indicates a positive correlation, whereas orange indicates a negative correlation.”, so 0.62 indicates a positive correlation. Correlation heatmap analysis by calculating the correlation coefficient between environmental factors and selected species, the obtained numerical matrix is visually displayed in the heatmap. The color change reflects the data information in the two-dimensional matrix or table, and the color depth represents the size of the data value. It can intuitively express the size of the data value with the defined color depth. Correlation r values are shown in different colors in Figure 6. The legend on the right is the color range of the different r values. The closer r value is to 1, the stronger the positive correlation between two variables. The closer the r value is to -1, the stronger the negative correlation between two variables. In addition, p value represents significance, and p < 0.05 indicates a significant correlation between two variables.

Point 12: Page 15 Line 495-516 low hemicellulose compounds in the solution “most discussion here explain about the relationship between bacteria and cellulose degradation. However, this study does not have analysis of cellulose, hemicellulose and lignin contents?

Response 12: Thank you for this suggestion. In many composting studies of agricultural waste, organic matter (OM) has been measured, while cellulose, lignin and hemicellulose content have been less measured. We have listed some references at the end. OM includes cellulose, lignin and hemicellulose, etc. According to our current research, we thought that changes in cellulose, lignin and hemicellulose might be the main reason of changes in OM content. We are doing further research on this problem.

  1. Paula, F.S.; Tatti, E.; Abram, F.; Wilson, J.; O'Flaherty, V. Stabilisation of spent mushroom substrate for application as a plant growth-promoting organic amendment. Environ. Manage. 2017, 196, 476-486. doi: 10.1016/j.jenvman.2017.03.038.
  2. Sun, Q.; Wu, D.; Zhang, Z.C.; Zhao, Y.; Xie, X.Y.; Wu, J.Q.; Lu, Q.; Wei, Z.M. Effect of cold-adapted microbial agent inoculation on enzyme activities during composting start-up at low temperature. Technol. 2017, 244, 635-640. doi: 10.1016/j.biortech.2017.08.010.
  3. Wang, S.P.; Wang, L.; Sun, Z.Y.; Wang, S.T.; Shen, C.H.; Tang, Y.Q.; Kida, K. Biochar addition reduces nitrogen loss and accelerates composting process by affecting the core microbial community during distilled grain waste composting. Technol. 2021, 337, 125492. doi: 10.1016/j.biortech.2021.125492.
  4. Li, J.B.; Wang, X.T.; Cong, C.; Wan, L.B.; Xu, Y.P.; Li, X.Y.; Hou, F.Q.; Wu, Y.Y.; Wang, L.L. Inoculation of cattle manure with microbial agents increases efciency and promotes maturity in composting. 3 Biotech. 2020, 10, 128. doi: 10.1007/s13205-020-2127-4.
  5. Li, X.N.; Wang, P.L.; Chu, S.Q.; Xu, Y.L.; Su, Y.L.; Wu, D.; Xie, B. Short-term biodrying achieves compost maturity and significantly reduces antibiotic resistance genes during semi-continuous food waste composting inoculated with mature compost. Hazard. Mater. 2022, 427, 127915. doi: 10.1016/j.jhazmat.2021.127915.

Reviewer 2 Report

Overall, this is a good work, performed using state of the art methods and equipment. The manuscript is well written and clearly presented. The conclusions are sound and well supported by the obtained results.

Given this, no corrections are needed.

Author Response

Dear Editors and Reviewers:

Thank you for inviting us to submit a revised draft of our manuscript entitled “Screening of cold-adapted strains and its effects on physicochemical properties and microbiota structure of mushroom residue composting”. We also appreciate the time and effort you and each of the reviewers have dedicated to providing insightful feedback on ways to strengthen our manuscript. Thus, it is with great pleasure that we resubmit our manuscript for further consideration. We have incorporated changes that reflect the detailed suggestions you have graciously provided. We also hope that our edits and the responses we provide below satisfactorily address all the issues and concerns you and the reviewers have noted. Revised portion are marked in red.

Reviewer 3 Report

Good work!! Minor revisions are suggested before being accepted for publication. They are highlighted in the file enclosed herewith by text tracking method.

Author Response

Dear Editors and Reviewers:

Thank you for inviting us to submit a revised draft of our manuscript entitled “Screening of cold-adapted strains and its effects on physicochemical properties and microbiota structure of mushroom residue composting”. We also appreciate the time and effort you and each of the reviewers have dedicated to providing insightful feedback on ways to strengthen our manuscript. Thus, it is with great pleasure that we resubmit our manuscript for further consideration. We have incorporated changes that reflect the detailed suggestions you have graciously provided. We also hope that our edits and the responses we provide below satisfactorily address all the issues and concerns you and the reviewers have noted. Revised portion are marked in red. The main corrections and responses are as follows:

Response to Reviewer 3 Comments

Point 1: Line 18 composite microbial agent (CMA) made of…

Response 1: Thank you for this suggestion. We have modified the sentence to “This study screened out a composite microbial agent (CMA) including Bacillus (B.) cereus QS7 and B. pumilus QM6 that could grow at 10–15℃ and investigated the effects of the CMA as an inoculant on the physicochemical parameters and microbial communities of compost made from mushroom residue mixed with sheep manure.” Please see Lines 17-21.

Point 2: Line 23 When ratio was...., compost increased....

Response 2: Thank you for this suggestion. We have modified the sentence to “When ratio of mushroom residue to sheep manure was at 9:1, compost increased the accumulation of nitrogen, and reduced the electrical conductivity (EC).” Please see Lines 25-26.

Point 3: Line 39 “demonstrated” and Line 29 “provided”

Response 3: Thank you for this suggestion. We have modified the sentence to “This study demonstrated that CMA improves the quality and efficiency of mushroom residue and sheep manure composting, and it provided evidence to improve the efficiency of low-temperature composting.” Please see Lines 33-35.

Point 4: Line 39 I suggest to read the following reference: https://doi.org/10.1016/j.scitotenv.2020.139840

Response 4: Thank you for this suggestion. We have added the article and have modified the sentence to “Composting is a common method for recycling agricultural waste, which can be converted into mature and stable end products and used as substrate for plant cultivation [3-6].” Please see Lines 43-46.

Point 5: Line 44 Why? Explain better.

Response 5: Thank you for this suggestion. We have modified the sentence to “However, composting requires a certain level of compost heap temperature. Low temperature conditions may cause composting to fail to start properly or to reach the high temperatures required to eliminate pathogenic microorganisms and harmful pollutants, posing technical challenges to the normal process of composting.” Please see Lines 49-53.

Point 6: Line 55 Quote.

Response 6: Thank you for this suggestion. We have already mentioned the reference number in Line 58 of the original manuscript. We have adjusted its position and have changed the sentence to “Lu et al [9] showed that inoculation with compound straw-decomposing microbial agents (a combination of cellulolytic microbes, Bacillus, and actinomycetes) accelerated the increase in compost pile temperature, promoted the degradation of total organic carbon (TOC), and increased the stability of the fungal network.”

Point 7: Line 230 of the strains QS7 and QM6.

Response 7: Thank you for this suggestion. We have modified the sentence to “Morphological, physiological and biochemical properties of the strains QS7 and QM6.” Please see Line 243.

Point 8: Line 239 was measured....how? and showed three....

Response 8: Thank you for this suggestion. We have modified the sentence to “The core temperature of the various compost piles were measured and showed three composting stages: heating up (days 0-4), high temperature (days 4-35), and cooling down (days 35-60).” Please see Lines 251-254.

Point 9: Line 251 delete "the"

Response 9: Thank you for this suggestion. We have modified the sentence to “Evolution of temperature during composting.” Please see Line 266.

Point 10: Line 351 move "at the genus level" after communities

Response 10: Thank you for this suggestion. We have modified the sentence to “Principal component analysis (PCA) of bacterial and fungi community on the genus level during fermentation.” Please see Lines 369-370.
